# Metal and Metal Oxide Nanoparticles in Caries Prevention: A Review

**DOI:** 10.3390/nano11123446

**Published:** 2021-12-20

**Authors:** Mohammed Zahedul Islam Nizami, Veena W. Xu, Iris X. Yin, Ollie Y. Yu, Chun-Hung Chu

**Affiliations:** Faculty of Dentistry, University of Hong Kong, Hong Kong 999077, China; nizami01@hku.hk (M.Z.I.N.); u3008489@connect.hku.hk (V.W.X.); irisxyin@hku.hk (I.X.Y.); ollieyu@hku.hk (O.Y.Y.)

**Keywords:** metal, nanoparticles, caries

## Abstract

Nanoparticles based on metal and metallic oxide have become a novel trend for dental use as they interfere with bacterial metabolism and prevent biofilm formation. Metal and metal oxide nanoparticles demonstrate significant antimicrobial activity by metal ion release, oxidative stress induction and non-oxidative mechanisms. Silver, zinc, titanium, copper, and magnesium ions have been used to develop metal and metal oxide nanoparticles. In addition, fluoride has been used to functionalise the metal and metal oxide nanoparticles. The fluoride-functionalised nanoparticles show fluoride-releasing properties that enhance apatite formation, promote remineralisation, and inhibit demineralisation of enamel and dentine. The particles’ nanoscopic size increases their surface-to-volume ratio and bioavailability. The increased surface area facilitates their mechanical bond with tooth tissue. Therefore, metal and metal oxide nanoparticles have been incorporated in dental materials to strengthen the mechanical properties of the materials and to prevent caries development. Another advantage of metal and metal oxide nanoparticles is their easily scalable production. The aim of this study is to provide an overview of the use of metal and metal oxide nanoparticles in caries prevention. The study reviews their effects on dental materials regarding antibacterial, remineralising, aesthetic, and mechanical properties.

## 1. Introduction

Dental decay, or caries, is a prevalent chronic disease that depends on multiple etiologic factors such as cariogenic microbes, host or tooth surface, substrate, and time. Although it is the hardest substance of human tissue, enamel can dissolve in acids. This dissolution or demineralisation of minerals causes the destruction of tooth structure and leads to dental caries (Figure 1a). Individuals are exposed to caries throughout their lifetime [1]. Dental caries is the most common condition among all oral diseases [2,3]. The World Health Organization has reported that dental caries is the fourth most expensive disease to treat, causing a significant global burden of disease [4]. Dental caries is not self-limiting, and without treatment it can advance, causing pain and infection until tooth loss [5]. However, caries is preventable. Preventive measures against dental caries have been remarkably improved in the last few decades. Several dental materials are employed to manage dental caries, including metals, ceramics, polymers, and hybrids. Contemporary nanotechnology has shown great attention in the development of anti-caries agents using nanoparticles.

Nanoparticles usually range in size from 1 to 100 nm. They may present in the form of atomic clusters, nanorods, dots, grains, fibres, films, or nanopores with high surface area. They have improved physicochemical properties compared to the conventional materials [6,7]. Nanoparticles exhibit antimicrobial, antiviral, and antifungal activities. In addition, nanoparticles may increase the mechanical properties, prevent crack propagation, and enhance fracture toughness of dental materials. Thus, applications of nanoparticles in dentistry have proliferated. Nanoparticles are considered useful in preventive dentistry, restorative dentistry, endodontics, implantology, prosthetic dentistry, oral cancers, and periodontology [8,9,10,11,12].

Current research outlines that metallic nanoparticles can inhibit dental caries by reducing biofilm formation and remineralising carious lesion (Figure 1b) [13,14]. Metallic nanoparticles stimulate biomineralisation by facilitating remineralisation of demineralised (carious) tooth tissues. Moreover, metallic nanoparticles are capable of overcoming challenges in diverse oral conditions due to their ion balance in oral fluid. Based on its potential benefits in various applications, researchers and clinicians have investigated several nano-formulations for caries prevention [15]. This review presents an overview of the development of metallic nanoparticles in prevention of dental caries.

## 2. Metal and Metal Oxide Nanoparticles Used in Caries Prevention

### 2.1. Silver Nanoparticles

Silver nanoparticles are broad-spectrum and non-resistant antimicrobial agents that can be used for caries prevention [16]. They attach to the outer cell membrane of bacteria due to the particles’ high surface area and alter the bacteria’s permeability and cell structure. Therefore, silver nanoparticles can kill bacterial cells effectively at minute concentration [17,18,19]. In vitro, in vivo, and clinical studies have reported use of silver nanoparticles in caries control. Studies have assessed various formulations of silver nanoparticles to inhibit cariogenic bacteria. Silver nanoparticles inhibit clinical isolate planktonic *Streptococcus mutans* and their mature biofilms [20]. In addition, they enhance the microhardness of tooth tissues with desirable antibacterial properties [21].

Silver nanoparticles incorporated in dental materials have been studied alongside silver nanoparticles alone. Silver nanoparticles in conventional sealants were found to enhance remineralisation [22]. Silver nanoparticle-coated orthodontic bracket showed efficacy in *S. mutans* inhibition and reduction in caries on the smooth enamel surface [23]. Compared to other cariostatic agents, silver nanoparticles showed a similar effect in caries prevention [24]. Silver nanoparticles incorporated into a poly(methyl methacrylate) or acrylic baseplate of a dental appliance can inhibit the planktonic growth and biofilm formation of cariogenic bacteria with desirable biocompatibility and mechanical properties [25,26].

Table 1 shows the properties of silver nanoparticles and their related products in caries prevention. Silver nanoparticles incorporated into acrylic plates showed strong antibacterial activity [27]. Silver nanoparticles in methacrylate imparted antibacterial properties without or with minimal change in mechanical properties [28]. Addition of silver nanoparticles in a hybrid composite resin enhanced the antibacterial effect against *Streptococcus* and *Lactobacillus* [29]. Silver nanoparticles in quaternary ammonium dimethacrylate reduced biofilm viability, metabolic activity, and acid production of cariogenic bacteria to prevent caries [30]. The silver nanoparticles showed a synergistic effect against microorganisms with minimal effect on the mechanical properties of the original cement when it was combined with amoxicillin and incorporated into glass ionomer cement restorative material [31].

Silver nanoparticles can be synergistically used with other nanoparticles. Silver nanoparticle-integrated amorphous calcium phosphate nanoparticles have ion-releasing properties that enhance their antibacterial and remineralising properties without altering their basic properties [32,33]. Reduced graphene oxide-silver nanoparticles were reported as a protective composite against enamel caries progression. The nanoparticles showed antimicrobial activity as well as reduction in enamel surface roughness while lowering the lesion depth and reducing mineral loss [34]. In another study, graphene oxide-silver-calcium fluoride nanoparticles demonstrated antimicrobial, cytocompatible, and sealing effects in cariogenic conditions [35].

Silver nanoparticles incorporated in sodium fluoride showed potential in remineralisation and caries prevention [36,37]. A clinical study demonstrated that silver nanoparticles with sodium fluoride solution inhibit dentine caries without staining [38]. An in vivo study also suggested that the fluoride varnish added with silver nanoparticles is effective in dental remineralisation [39].

### 2.2. Zinc Nanoparticles

Zinc nanoparticles can inhibit *S. mutans*, reduce plaque formation and facilitate remineralisation [40]. Zinc oxide nanoparticles are more biocompatible than zinc nanoparticles [41]. Zinc oxide nanoparticles have better antimicrobial activity than other zinc nanoparticles [42,43]. In addition, zinc oxide nanoparticles inhibit growth of *S. mutans* [44]. The nanoparticles can be added to restorative materials to prevent caries. Zinc oxide nanoparticles can be incorporated into conventional glass without altering basic mechanics [45]. Moreover, zinc oxide nanoparticles conjugated composite resin exhibited antibacterial activity against *S. mutans* [46,47]. Table 2 gives a brief overview of the application of zinc nanoparticles in caries prevention.

Zinc oxide nanocomposites and chlorhexidine-containing composites showed similar antibacterial effects [48]. However, a study reported that zinc oxide did not promote antimicrobial activity of composite resin against *S. mutans* [49]. As with zinc oxide’s antibacterial effect, studies reported conflicting results of zinc oxide nanoparticles on mechanical properties of composite resin [50]. A study reported that the addition of zinc oxide nanoparticles to flowable composite resin did not affect mechanical properties of commercial composite resin [51]. Another study found that the application of zinc oxide nanoparticles into composite resin affected the chemical and mechanical properties [52].

Zinc oxide nanoparticles can be incorporated with other metal nanoparticles for caries prevention. The addition of zinc oxide and copper nanoparticles into universal adhesive can enhance the antimicrobial activity against *S. mutans* and have anti-matrix metalloproteinase properties without changing mechanical properties [53]. Likewise, fluoride-containing zinc oxide and copper oxide nanoparticles were reported to exhibit antibacterial action, enzyme inhibition, and bio-mineralisation in cariogenic conditions [54]. Moreover, silver-zinc oxide nanocomposite demonstrated strong antimicrobial activity against *S. mutans* [55]. The addition of silver-zinc oxide nanoparticles in the composite resin showed significant inhibitory effect on *S. mutans* biofilm without altering compressive strength [56].

Researchers studied the effect of zinc oxide nanoparticles combined with organic compounds on caries prevention. The combined chitosan hydrogel and zinc oxide-zeolite nanocomposite was non-cytotoxic to human gingival fibroblast cells and reduced the formation of *S. mutans* biofilm [57]. A study reported that cellulose nanocrystal-zinc oxide nanohybrids in dental resin possess improved compressive strength, flexural modulus, and antibacterial properties [58]. In addition, zinc oxide-decorated graphene nanocomposite demonstrated an inhibition effect on *S. mutans* [59,60]. In conclusion, the zinc nanoparticles without other components are not effective in inhibiting bacteria. Researchers might focus on exploring novel zinc nanoparticles and their composite or hybrid formulation in caries prevention.

### 2.3. Titanium Nanoparticles

Titanium nanoparticles show potential in dental applications for their superior biocompatibility, bioactivity, and wide-ranging antimicrobial activity. Titanium nanoparticles can be used as an antibacterial agent against cariogenic bacteria and biofilm as they can initiate small pores in bacterial cell walls, leading to broader permeability and cell death [61]. In addition, titanium nanoparticles possess high stability, photocatalytic activity, and reusability, which make titanium nanoparticles a potential alternative in caries prevention [62,63].

Titanium dioxide nanoparticles is one of the titanium nanoparticles widely studied in caries prevention. Glass ionomer cement containing titanium nanoparticles is a promising restorative material for caries prevention [64]. Incorporation of titanium nanoparticles into restorative glass ionomer cement significantly improved antibacterial activity, microhardness, flexural, and compressive strength without decreasing adhesion to enamel and dentine [65]. Nitrogen-doped titanium nanoparticles displayed higher light absorption levels when compared to undoped titanium oxide. Thus, experimental adhesives containing nitrogen-doped titanium nanoparticles demonstrated superior antibacterial efficacy in dark conditions [66]. Table 3 shows uses of titanium nanoparticles for caries prevention. Researchers might utilize realistic ideas and limitations for their future study design.

### 2.4. Calcium Nanoparticles

Hydroxyapatite (Ca_5_(PO_4_)_3_(OH)) composed of calcium and phosphate is the main inorganic component of teeth. A balanced level of calcium and phosphate is crucial to prevent caries [67]. However, the clinical application of calcium and phosphate ions has not been successful in the last few decades. Insoluble calcium phosphates are not easy to apply in dental practices. Soluble calcium and phosphate ions can only be used at very low concentrations due to the inherent insolubility of calcium phosphates. Furthermore, soluble calcium and phosphate ions cannot extensively integrate with dental plaque or deposit on the tooth surface. Thus, the bioavailability of calcium and phosphate ions is always limited in the remineralisation process. However, recent advancements in nanotechnology have displayed several kinds of calcium nanoparticles in application of caries prevention. Moreover, calcium phosphate nanoparticles were used to develop a new composite with high stress-bearing and caries-preventing properties [68].

Amorphous calcium phosphate nanoparticles are capable of recharging and re-releasing calcium and phosphate ions. Thus, amorphous calcium phosphate nanoparticles can promote remineralisation through long-term and sustained release of calcium and phosphate ions [69,70]. Amorphous calcium phosphate nanoparticles can be added to orthodontic cement to avoid white spot lesions during orthodontic treatments due to their ability to inhibit caries and remineralise lesions [71]. Adhesive containing amorphous calcium phosphate nanoparticles can remineralise dentine lesions in a biofilm model, develop a strong bond interface, inhibit secondary caries, and extend the longevity of the restoration [72].

Amorphous calcium phosphate nanoparticles can be combined with many other organic agents to realise the remineralising and antibacterial effects. Quaternary ammonium methacrylate-amorphous calcium phosphate nanoparticles were used in various studies for their remineralisation capability, inhibition of biofilm growth and lactic acid production and increase in dentine bond strength [73,74,75,76,77,78]. A composite containing quaternary ammonium methacrylate-amorphous calcium phosphate nanoparticles can facilitate the healing of the dentine–pulp complex and dentine formation [79]. A study reported that 2-methacryloxylethyl dodecyl methyl ammonium bromide and amorphous calcium phosphate nanoparticles exhibited a strong reduction in demineralisation and inhibition of biofilm formation without altering the shear bond strength of the composite [80].

Combined salivary statherin-protein-inspired polyamidoamine dendrimer and amorphous calcium phosphate nanoparticles facilitate enamel remineralisation in artificial caries model [81]. An adhesive resin containing triple agents of shells containing triethylene glycol dimethacrylate, quaternary ammonium methacrylate, and amorphous calcium phosphate nanoparticles possessed antimicrobial and remineralising properties for prevention of secondary caries [82]. Other calcium-containing nanoparticles were studied recently; a dentifrice containing nano-carbonated apatite and fluoride was reported as an effective remineralising agent to prevent incipient caries lesions [83]. Another study used nano-calcium carbonates to remineralise tooth enamel and showed nano-calcium carbonates potential to remineralise initial enamel lesions [84].

Fluoride can combine with calcium and phosphate and form fluorapatite (Ca_5_(PO_4_)_3_F), which is more resistant to acid attack than hydroxyapatite. The presence of fluoride can accelerate remineralisation of carious lesions and hinder demineralisation of enamel and dentine [85]. Thus, calcium fluoride nanoparticles were developed as anti-caries agents due to their increased level of labile fluoride concentration in oral fluid. An in vitro study revealed calcium fluoride nanoparticles can substantially reduce exopolysaccharide production and inhibit biofilm formation [86]. Furthermore, calcium fluoride nanoparticles can enhance the effect of remineralisation [87]. Calcium fluoride nanoparticles were able to decrease caries in treated rat groups [88]. In addition, calcium fluoride nanoparticles were incorporated in a nanocomposite to obtain strong mechanical durability, high strength, and high fluoride release capability [89]. In addition, composite incorporating calcium fluoride nanoparticles increased ion release of fluoride and calcium [90]. Table 4 shows a summary of calcium nanoparticle application in caries prevention.

### 2.5. Copper Nanoparticles

Copper nanoparticles can inhibit growth and colonisation of *S. mutans* on the surface of tooth root and prevent root caries [91]. Copper oxide nanoparticles cost less than silver nanoparticles do. They have desirable physical properties with high surface area and crystalline structure. A review concluded that copper oxide nanoparticles are bactericidal against cariogenic bacteria [92]. In addition, the antimicrobial activity of copper oxide nanoparticles is dose dependent [93]. Copper oxide nanoparticles can be easily mixed into polymers to provide composites with unique physio-chemical properties.

As copper oxide nanoparticles are antimicrobial without altering shear bond strength, they can be incorporated into dental adhesive to prevent early or carious white spot lesions [94,95]. Other copper nanoparticles, including copper iodide nanoparticles and copper fluoride nanoparticles, also exhibit antibacterial effects against *S. mutans* [96,97]. Table 5 outlines the use of copper nanoparticles in caries prevention. Researchers and clinicians might consider copper nanoparticles an alternative in further investigations as a relatively low-cost metal nanoparticle in clinical settings.

### 2.6. Magnesium Nanoparticles

The caries process involves demineralisation of hydroxyapatite due to acid attack [98,99,100]. Thus, alkaline nanoparticles could be an alternative for caries prevention. Magnesium is an alkaline metal that constitutes approximately 0.5% of enamel and 1% of dentine [101]. A study showed that adequate levels of serum magnesium could reduce the progression and development of dental caries through release of magnesium ions [102]. Magnesium oxide-nanoparticle-modified glass ionomer cement showed significant antibacterial and biofilm activity against cariogenic bacteria [103]. A limited study reported using magnesium nanoparticles for caries prevention. Magnesium expresses both significant [104] and non-significant relations with tooth decay [105].

## 3. Applications of Nanotechnology in Dentistry

Nanotechnology has broad applications in dentistry, in particular for dental materials. Further studies should be conducted to investigate the durability and sustainability of these metal nanoparticles and their conjugates. In addition, studies should examine the bioavailability and biocompatibility, binding energy, and binding potential to tooth tissues, the integrity and mechanics with tooth tissues, and the aesthetic outcome. Clinical trials should be conducted to validate their effectiveness in caries prevention.

Studies of nanoparticles applications in dentistry often followed by confocal Raman spectroscopy. Confocal Raman microscopy is an enhanced system to evaluate depth and buried structures of materials. This label-free method offers essential information about the effect of metal or metal-oxide nanoparticles. Researchers are using bio-inspired nanoparticles to promote remineralisation and prevent caries development. The addition of various metallic nanoparticles in dentifrices and mouth-rinsing solutions has proven to promote anti-caries properties. Moreover, metallic nanoparticles in dental polishing agents and filling materials are also found to prevent caries [106,107]. Studies have demonstrated that metallic nanoparticles, alone or in composites, have antibacterial and remineralising properties [108,109]. They may also be used to improve aesthetic and mechanical properties of dental materials. Bioactive nanoparticles can be incorporated into dental products for daily oral care. Dentifrices and mouth rinse solutions containing nanoparticles exhibit antimicrobial, remineralising, and anti-inflammatory properties. Saliva contains ions and proteins that may form nanoparticles–ion–protein complexes with nanoparticles and precipitate onto the tooth surfaces.

## 4. Conclusions

Researchers have been developing innovative metal nanoparticles for dental use. Silver, zinc, titanium, copper, and magnesium ions have been used to develop metal and metal oxide nanoparticles. With advances in research and development of nanotechnology, nanomaterials with improved physiochemical, antibacterial, or remineralising properties have been developed. The research into vast earth metal and metallic oxides could be a promising strategy for management of dental caries, which affects 2.4 billion people, or one-third of the global population.

## Figures and Tables

**Figure 1 nanomaterials-11-03446-f001:**
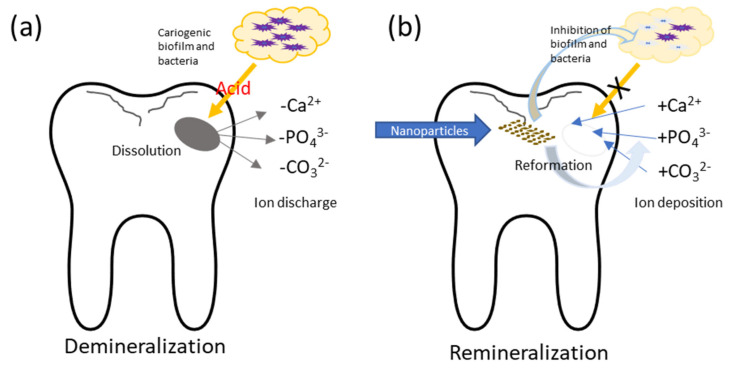
Mechanism of (**a**) demineralisation and (**b**) remineralisation of the tooth surface.

**Table 1 nanomaterials-11-03446-t001:** Properties of silver nanoparticles and their related products in caries prevention.

** *Silver nanoparticles:* ** Inhibiting growth of *S. mutans* [16,20,23,24,27,29], *S. sobrinus* [16], *S. sanguinis* [16], *S. gordonii* [16], *S. oralis* [16], *L. acidophilus* [29]; Inhibiting demineralisation [22]; Promoting remineralisation [21,22]; Synergistically inhibiting growth of *S. mutans* and *S. aureus* with amoxicillin [31].
** *Poly(methyl methacrylate)—silver nanoparticles:* ** Inhibiting growth of *S. mutans*, *S. sobrinus*, *L. acidophilus*, and *L. casei* [25].
** *Poly(methyl methacrylate)—cellulose nanocrystal—silver nanoparticles:* ** Inhibiting growth of *S. aureus* [26]; Improving mechanical property [26]; Being biocompatible [26].
** *Amorphous calcium phosphate—quaternary ammonium dimethacrylate—silver nanoparticles:* ** Inhibiting growth of *S. mutans* [30]; Reducing production of lactic acid from *S. mutans* biofilm [30].
** *Amorphous calcium phosphate nanoparticles—silver nanoparticles:* ** Releasing silver, calcium and phosphorus ions [32]; Non-decreasing mechanical strength [32]; Inhibiting growth of *S. mutans* [32,33]; Promoting remineralisation [33].
** *Reduced graphene oxide—silver nanoparticles:* ** Inhibiting growth of *S. mutans* [34]; Promoting remineralisation [34].
** *Graphene oxide-silver-calcium fluoride nanoparticles:* ** Inhibiting growth of *S. mutans* [35]; Sealing orifices of dentinal tubules [35]; Non-discolouring teeth [35].
** *Nanosilver fluoride:* ** Inhibiting demineralisation [22]; Promoting remineralisation [21,22]; Non-discolouring teeth [36,37,38]; Arresting dentine caries [38].

**Table 2 nanomaterials-11-03446-t002:** Properties of zinc nanoparticle and their related products in caries prevention.

** *Zinc oxide nanoparticles:* ** Inhibiting growth of *S. mutans* [44,50,51,52], *S. aureus* [46], *S. sobrinus* [48], and oral biofilm [52]; Uncompromising mechanical properties [51,52] and bond strength [59].
** *Glass ionomer cement—zinc oxide nanoparticles:* ** Increasing minimal mechanical property [45].
** *Copper—zinc oxide nanoparticles:* ** Improving integrity of the hybrid layer on caries-affected dentine [52]; Inhibiting growth of *S. mutans* [53]; Promoting anti-matrix metalloproteinase activities [53].
** *Copper oxide-fluoride-zinc oxide nanoparticles:* ** Inhibiting growth of *S. mutans* [54]; Promoting remineralisation [54].
** *Silver—zinc oxide nanoparticles:* ** Inhibiting growth of *S. mutans* [55,56]; Uncompromising compressive strength [56].
** *Chitosan hydrogel—zinc oxide nanoparticles:* ** Inhibiting growth of *S. mutans* [57]; Showing non-cytotoxic effect on human gingival fibroblast cells [57].
** *Cellulose nanocrystal—zinc oxide nanoparticles:* ** Inhibiting growth of *S. mutans* [58]; Promoting mechanical properties [58].
** *Graphene—zinc oxide nanoparticles:* ** Inhibiting growth of *S. mutans* [59,60].

**Table 3 nanomaterials-11-03446-t003:** Properties of titanium nanoparticles and their related products in caries prevention.

** *Titanium nanoparticles:* ** Inhibiting growth of *S. mutans* [64,65]; Promoting mechanical properties [65]; Uncompromising bond strength [66].
** *Nitrogen-doped titanium nanoparticles:* ** Inhibiting growth of *S. mutans* [64,65].

**Table 4 nanomaterials-11-03446-t004:** Properties of calcium nanoparticles and their related products in caries prevention.

** *Amorphous calcium phosphate nanoparticles:* ** Releasing calcium and phosphorus [68,70,71]; Neutralising acid [69]; Promoting remineralisation [69]; Inhibiting demineralisation [70]; Inhibiting growth of *S. mutans* [71]; Reducing production of lactic acid from biofilm [71].
** *Quaternary ammonium methacrylate—amorphous calcium phosphate nanoparticles:* ** Inhibiting growth of *S. mutans* [72,73], multi-species biofilm with *S. mutans*, *L. acidophilus*, and *C. albicans* [73] and dental plaque microcosm biofilm [72,73,75]; Reducing production of lactic acid from biofilm [72,73,74]; Releasing calcium and phosphorus [73,74,75,76,77]; Promoting remineralisation [72,78]; Inhibiting demineralisation [74,77]; Uncompromising bond strength [75].
** *2-methacryloxylethyl dodecyl methyl ammonium bromide—amorphous calcium phosphate nanoparticles:* ** Releasing calcium and phosphorus [79]; Increasing enamel hardness [79].
** *Salivary statherin protein-inspired poly(amidoamine) dendrimer—amorphous calcium phosphate nanoparticles:* ** Releasing calcium and phosphorus [80]; Promoting remineralisation [80].
** *Triethylene glycol dimethacrylate-quaternary ammonium methacrylate-amorphous calcium phosphate nanoparticles:* ** Inhibiting growth of *S. mutans* [81]; Promoting remineralisation [81]; Uncompromising bond strength [81].
** *Nano carbonated apatites:* ** Promoting remineralisation [82].
** *Nano calcium carbonates:* ** Releasing calcium [83]; Promoting remineralisation [83].
** *Calcium phosphate nanoparticles:* ** Releasing calcium and phosphorus [84]; Promoting stress-bearing ability [84].
** *Calcium fluoride nanoparticles:* ** Inhibiting growth of *S. mutans* [88]; Reducing acid production from biofilm [88]; Releasing fluoride [86,89]; Promoting remineralisation [86]; Inhibiting dentine permeability [86].
** *Quaternary ammonium methacrylate—calcium fluoride nanoparticles:* ** Inhibiting growth of *S. mutans* [87,90]; Releasing calcium and fluoride [87,90]; Promoting remineralisation [90].

**Table 5 nanomaterials-11-03446-t005:** Properties of copper nanoparticle and their related products in caries prevention.

** *Copper nanoparticles:* ** Inhibiting growth of *S. mutans* [91].
** *Copper oxide nanoparticles:* ** Inhibiting growth of *S. mutans* [92,94,95], *L. casei* [92,94], and *L. acidophilus* [92]; Inhibiting growth of *C. albicans*, *C. krusei*, and *C. glabrata* [94]; Inhibiting demineralisation [94]; Uncompromising mechanical properties [95].
** *Copper iodide nanoparticles:* ** Inhibiting growth of *S. mutans* [96]; Uncompromising bond strength [96].
** *Copper fluoride nanoparticles:* ** Inhibiting growth of *S. mutans* [97].

## Data Availability

Not applicable.

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
