# Peer review of "Metal and Metal Oxide Nanoparticles in Caries Prevention: A Review"

_nanomaterials, 2021, doi:10.3390/nano11123446_

Round 1

Reviewer 1 Report

Nanomaterials 1502666

In this review, the authors show the recompilation of nanoparticles based on metal and metallic oxide for its application in a novel trend, use because they interference dental with bacterial metabolism and present biofilm formation. The ions used are, silver, zinc, titanium, copper, and magnesium have been used to functionalize the metal and metal oxide nanoparticles provide remineralization and inhibit demineralization.

Metal and metal oxide nanoparticles have been incorporated in dental materials to strengthen the mechanical properties and prevent the caries. The study reviews the effects on dentals materials regarding antibacterial, remineralizing and mechanical properties.

This paper is well documented, and the tables contain a lot of information, also is highly schematized.

This paper can be published in Nanomaterials, but with some changes,

Comment

  • The zinc nanoparticles without other components are not very effective. Comment with major detail this part (2.2)
  • Complete the conclusions

Other comment

  • Abstract, Line 4, change zinc titanium for zinc, titanium
  • Page 3, Line 88, change prevention.Silver for prevention. Silver
  • Page 4, Line 120, delete “are”
  • Page 4, Line 121, delete “have”
  • Page 5, Line 161, change their for to have
  • Page 5, Line 173, change microhardness and flexural for microhardness, flexural…Revise all manuscript
  • Page 6, Line 185, add Ca5(PO4)3(OH)
  • Page 6, Line 230, add Ca5(PO4)3F
  • Page 6, Line 226-228, change the “Moreover, ……preventing properties” in paragraph 1 of the Part 2.4
  • Page 9, Line 297, change zinc titanium for zinc, titanium
  • Revise the references: [4] [38] [48] [66] [85]

Reviewer 2 Report

In this work authors provide an overview of the use of metal and metal oxide nanoparticles in caries prevention. The study reviews their effects on dental materials regarding antibacterial, remineralising, aesthetic and mechanical properties. The work is interesting and surely will earn wide interest justifying publication in Nanomaterials. The references are up to date and comprehensive. I suggest publication after considering the following minor remark:

The nanoparticles applications in dentistry often followed by confocal Raman spectroscopy. I suggest authors to include this field into their overview since this label-free method offers several information about the effect of metal or metal-oxide nanoparticles.
